# Unsupervised Representation Learning to Aid Semi-Supervised Meta Learning

## Abstract

Few-shot learning or meta-learning leverages the data scarcity problem in machine learning. Traditionally, training data requires a multitude of samples and labeling for supervised learning. To address this issue, we propose a one-shot unsupervised meta-learning to learn the latent representation of the training samples. We use augmented samples as the query set during the training phase of the unsupervised meta-learning. A temperature-scaled cross-entropy loss is used in the inner loop of meta-learning to prevent overfitting during unsupervised learning. The learned parameters from this step are applied to the targeted supervised meta-learning in a transfer-learning fashion for initialization and fast adaptation with improved accuracy. The proposed method is model agnostic and can aid any meta-learning model to improve accuracy. We use model agnostic meta-learning (MAML) and relation network (RN) on Omniglot and mini-Imagenet datasets to demonstrate the performance of the proposed method. Furthermore, a meta-learning model with the proposed initialization can achieve satisfactory accuracy with significantly fewer training samples.

## 1 Introduction

Meta-learning is a relatively new branch of machine learning that deals with learning to learn problems **?** with only a few samples. Traditional machine learning algorithms require massive datasets to reach their peak performance. Nevertheless, these algorithms suffer if the test domain slightly deviates from the training domain. Furthermore, if a new class is introduced, it requires training from scratch again. On the other hand, human learning is far more advanced as they can learn from only a few samples and distinguish a new class without seeing many samples. It is because humans use their previous memory when learning a new task. Meta-learning mimics the process of human learning and tries to bridge the gap between machine learning and human learning **?**.

Almost all meta-learning algorithms **????** deal with a task or episode generation during the training phase to learn to use this knowledge during the testing phase for being able to distinguish from a few samples. This phenomenon is defined as learning to learn, and both the training and testing phase have samples that are called support and query sets **?**, respectively. The support set is used for learning the class representation, and the query set is applied for inference. All meta-learning algorithms are built on this fundamental strategy. Support and query sets are generated in batches (also known as episodes in meta-learning lingo) by drawing samples from the training data. A One hot encoded pseudo labels are added to the classes in the episodes. Exact class labelling is not essential at this stage because, during the training time, meta-learning algorithms only try to learn to perform testing on some new classes never seen before. This motivates our study to use random training samples for support sets from the pool of the training data and generate query sets using the augmented training samples. This pseudo-labeling helps the classifier learn some feature representations from the dataset without going through the time-consuming manual labeling process.

Our proposed method uses specific image augmentation techniques to generate the training episodes. First, we lose all the labels and class information from our data pool. Then we randomly draw samples from the pool to generate our support sets and do image augmentation on the support sets to generate our query sets. Technically, it works for datasets like Omniglot **?** and mini-Imagenet **?**

or larger datasets because they contain a multitude of samples and classes, and the probability of drawing from the same class, is much lower.

Our method contains two steps of training. First, the fully unsupervised training to learn the latent representations of the dataset. We use the labeled test sets to observe the performance during this time. The meta-learning algorithm achieves some accuracy during unsupervised representation learning, although not as good as supervised learning. Later, these learned parameters are used to initialize the final supervised meta-learning and to boost the performance. Therefore, in the second step of meta-learning, we initialize with the learned parameters from the unsupervised learning model instead of random initialization. Thus, the whole process becomes a semi-supervised meta-learning **?**.

For an effective augmentation technique, we followed the suggestion from the SimCLR **?** with a few additional augmentations to increase the effectiveness. Our proposed method is model agnostic and can be applied to any meta-learning model. We used two prominent meta-learning architectures, model agnostic meta-learning (MAML) **?** and relation network (RN) **?**, to test our hypothesis. We also modified a part of the MAML network architecture by adding temperature **?** to the SoftMax activation function in the inner loop of MAML to reduce overfitting during the unsupervised training. We did not modify the RN architecture but used our hyperparameters and architecture to obtain higher accuracy than reported in the original paper. Our proposed method can enhance the accuracy of any state-of-the-art meta-learning model, as proved in the experiments of this study.

Our contributions to this work are listed below:

- We proposed a more effective data augmentation technique to generate query sets by combining techniques from SimCLR and our additional steps.
- We used a temperature-scaled SoftMax in the inner steps of MAML to reduce overfitting during meta-training. Our implementation of RN surpasses the accuracy of the original RN.
- We replaced random initialization of meta-learning with unsupervised representation learning for inherent feature learning that does not require extensive data labeling. After transferring the parameters from unsupervised learning, we applied supervised meta-learning to achieve improved accuracy.
- We showed that our two steps meta-learning is model agnostic and improves the accuracy of any existing meta-learning model. We also experimented with partially labeled data and found that the classifier loses insignificant accuracy when trained with our method.

## 2 RELATED WORK

Meta-learning **?** has many practical applications, such as self-driving cars, face recognition, and computer vision. Although the core motivation of meta-learning is to classify with a few samples, training the model still requires a lot of labeled samples. This popularized the use of data augmentation in meta-learning. Yao et al. **?** proposed two task augmentation methods, called MetaMix and channel shuffle. MetaMix linearly combines features and labels of samples from both the support and query sets. Channel shuffle randomly replaces a subset of their channels with the corresponding ones from a different class. Experimental analysis showed that their method effectively reduces overfitting in meta-learning. Rajendran et al. **?** introduced an information-theoretic framework of meta-augmentation for better generalization by adding randomness, which discourages the base learner and model from learning unimportant features. Nevertheless, all these methods are supervised learning and still need the labeling of a large number of samples.

Hsu et al. proposed one of the earliest unsupervised meta-learning algorithm called CACTUs **?** which assigns pseudo level to the remaining unlabelled datasets using a nearest neighbor approach. It is an iterative process where the pseudo-labels are incorporated into the clustering and adaptation steps leading to an improved accuracy. Nevertheless, the proposed method requires additional steps such as embedding learning algorithm and $k$-means clustering **?** for the purpose of pseudo label generation. These extra steps make the algorithm computationally expensive. Moreover, the authors did not extend the idea to semi-supervised learning. Therefore, the method cannot match the accuracy of a supervised learning.

Khodadadeh et al. **?** proposed UMTRA, an algorithm that performs unsupervised, model-agnostic meta-learning for classification tasks. They used augmented query samples for the unsupervised

classification of MAML. However, their proposed method is fully unsupervised and ultimately achieves much lower accuracy than supervised meta-learning. Chen et al. **?** proposed SimCLR that investigates the most effective data augmentation for semi-supervised learning. They used a normalized temperature-scaled cross-entropy loss to achieve better generalization during the unsupervised representation learning. The two aforementioned pieces of research heavily influenced our proposed work to develop a semi-supervised meta-learning that utilizes the power of unsupervised representation learning and meta-transfer learning.

There are several state-of-the-art meta-learning models popular in the research community. MAML **?** is one of the pioneers of deep meta-learning models. MAML tries to find the optimal parameters over the task embeddings for fast adaptation. The family of MAML contains several popular and almost similar classifiers, namely, Reptile **?**, Meta-SGD **?**, LEO **?**. Another popular model is called Prototypical network **?**, which learns a metric space in which classification can be performed by computing distances to prototype representations of each class. This network obtains higher accuracy than many of its predecessors. RN **?** came out right after the Prototypical network, which surpassed the accuracy of the Prototypical network in most cases. Our study obtained promising outputs using a modified MAML for unsupervised learning and additionally uses RN to show its model-agnostic ability.

## 3 PROPOSED METHOD

### 3.1 STEP 1: UNSUPERVISED LEARNING

**Data Preparation:** To incorporate representation learning with meta-learning, we first take the entire or partial dataset without any label information. An effective way to learn the representation is to use both the labeled and unlabelled data. This ensures that the classifiers learn all the inherent representation in a semi-supervised way.

First, we draw the samples $x_{i,j}$ from the data pool of $X_N$ where $i, j$ are the number of shots and the number of ways, respectively, considered in the unsupervised learning and $N$ is the total number of unlabelled samples. We only design $n$-way ($n$ is the number of ways or classes), 1-shot support sets because each sample in the support set is drawn randomly, and we cannot randomly add more same-class support samples to that set. However, we can apply data augmentation for the query set to generate multiple query samples of the same class. But is generating more query samples more effective? We answer that question in the later part of this research.

The exact labeling in meta-training episodes is not crucial. Therefore, after generating the training episodes, we randomly assign labeled values $y_{i,j}$ to each class of the support sets, where $j$ is the number of ways generated as $\{c_0, c_1, ..., c_{j-1}\}$ and one-hot encoded later. We initialize the random initialization parameter for the unsupervised classifier, $\theta$. We randomly draw the support sets for each task episode and generate the randomly generated support labels. To generate the query set, we pass each sample of the support set through a data augmentation function $f(A)$ and similarly generate the pseudo labels. Ultimately, we use the regular supervised meta-learning learning test setup to examine the classifier's performance.

**Deep Dive into Support-Query Set Generation:** We intuitively know that when we draw a few samples from a large pool of data, more than one sample belonging to the same class is low. Therefore, we must ensure that $n << c$ where $n$ is the number of ways (or the drawn samples since we only apply 1-shot learning) and $c$ is the total number of classes. Nevertheless, we need to mathematically compute the probability of getting unique samples in each class for the datasets used in this study.

We use two different datasets, Omniglot and mini-Imagenet. The prior one has less number of samples in each class than the total number of classes. Therefore, it is most likely that all drawn samples will originate from different classes. The latter has more samples ($600$) in each class than the total number of classes. Therefore, the probability of originating from different classes would be slightly lower. Nevertheless, we have an equal number of samples in both datasets, $m$ for each class. Now, we can calculate the probability of the samples belonging to different classes as follows:

$$P = \frac{c! \cdot m^n (c \cdot m - n)}{(c - n)! \cdot (c \cdot m)!} \tag{1}$$

145 Using the aforementioned formula, the probabilities of 5-way 1-shot classification for the Omniglot
146 (1200 classes) and mini-Imagenet (64 classes) are 99.21% and 85.23%, respectively.

147 Effective data augmentation is important in this research to generate the query sample. We follow the
148 suggestion from the SimCLR **?** and combine it with other methods to make it more effective for the
149 RGB image classification (mini-Imagenet). SimCLR paper elaborates on the effectiveness of data
150 augmentation and choosing the proper augmentation function, which motivates us to follow their
151 method. They suggested the most effective combination of Gaussian blur, random crop, and random
152 color distortion. We added horizontal flip and random color invert (50% probability) with these three
153 methods as we found that it reduces overfitting and improves accuracy. On the other hand, for the
154 grayscale Omniglot dataset, we only use random affine transform because we found that both the
155 support and query samples are very similar, and a hard augmentation hurts the performance.

156 **Classifiers:** Our proposed method is model agnostic and can be applied to any model. In this paper,
157 we use two meta-learning models, MAML and RN, to demonstrate the performance on different
158 architectures. We find that for MAML, the classifier trained on RGB samples (mini-Imagenet in our
159 case) has a severe overfitting issue using the regular classifier. This is because the augmented query
160 samples are similar to the original support samples. Therefore, the classifier learns very little during
161 the training phase. We solve this problem by using a temperature-scaled SoftMax activation function
162 only in the inner loop of MAML. The temperature term makes the classifier less confident of the
163 support set samples, and thus the classifier can learn more information from the subtle differences.
164 The mathematical expression for temperature-scaled SoftMax is as follows:

$$\frac{\exp(z_i/T)}{\sum_{k=0}^{j-1} \exp(z_k/T)} \tag{2}$$

165 where the scaling is accomplished by dividing the logits of SoftMax by a value $T$, known as
166 temperature. $j$ is the number of ways, and $z_i$, $z_k$ represent the $i^{th}$, $k^{th}$ input to the SoftMax,
167 respectively.

168 We found RN performing counter effectively when using a temperature-scaled SoftMax. We instead
169 used our own set of hyperparameters that led to more improved accuracy than the RN in the original
170 paper.

171 After training the unsupervised learning algorithm, we save the weights and biases to perform
172 semi-supervised meta learning. Therefore, in the classifier of step-2, instead of randomly initialized
173 parameters, $\theta$, we used the transferred parameters, $\theta^*$. Then, we perform the regular meta-learning
174 for fine-tuning and improved accuracy.

## 3.2  STEP 2: SEMI-SUPERVISED META LEARNING (SSML)

176 In this step, we apply SSML on the regular meta-learning settings but initialize the weights and biases
177 from the first classifier. First, let us talk briefly about the two classifiers, MAML and RN.

178 **MAML:** MAML tries to find the optimal parameters $\theta$ derived from a few parametric models $f_\theta$.
179 In MAML, we generate the episodes from the data distribution such as $\tau_i = (D^{tr}, D^{val})$. We use
180 the gradient update to update the initialize parameter $\theta$ to $\theta'_i$ across tasks sampled from $p(\tau)$ and is
181 obtained as follows:

$$\theta'_i = \theta - \alpha \nabla_\theta \mathcal{L}_{\tau_i}(f_\theta) \tag{3}$$

182 where $\alpha$ is the learning rate of the meta-inner loop, and $\mathcal{L}$ is the loss function. In the outer loop of
183 meta-learning, the optimization is performed across tasks via stochastic gradient descent (SGD) to
184 update the $\theta$. It is obtained as follows:

$$\theta \leftarrow \theta - \beta \nabla_\theta \sum_{\tau_i \sim p(\tau)} \mathcal{L}_{\tau_i}(f_{\theta^i}) \tag{4}$$

185 where $\beta$ is the learning rate of the meta-outer loop.

186 **RN:** The main two components of RN are a feature extractor and a relation module. The feature
187 extractor concatenates the features from the support sets, and the query sets as $f_\varphi(x_i)$ and $f_\varphi(x_j)$
188 through a function $\mathbb{C}(f_\varphi(x_i), f_\varphi(x_j))$.

The combined features are passed through the relation module to obtain their relation score. It is passed through a Sigmoid activation function to obtain the score in a range between 0 to 1. The equation for that is provided below:

$$r_{i,j} = g_\phi(\mathbb{C}(f_\varphi(x_i), f_\varphi(x_j))) \tag{5}$$

To create the final output, the relation network's output can also be subjected to extra processing by layers, such as a fully connected neural network. Because of this, the relation network is an adaptable architecture that may be used for various applications. A mean-square-error (MSE) loss function is used to update the network using gradient descent.

$$\varphi, \phi \leftarrow \arg\min \sum_{i=1}^{m} \sum_{j=1}^{n} (r_{i,j} - \mathbb{1}(y_i == y_j))^2 \tag{6}$$

**Overall Summary:** The overall method is summarized in this sector with a diagram for better understanding. Figure 1 depicts the steps of the proposed method. We generate the training episodes from the unlabeled samples. Here, the NT-Xent loss **?** (temperature-scaled SoftMax) is only applied on the MAML for the mini-Imagenet dataset. After training the initial model, we save the parameters and transfer them to the final model for improved performance. Moreover, the pseudo-code for our proposed method is provided in Algorithm 1.

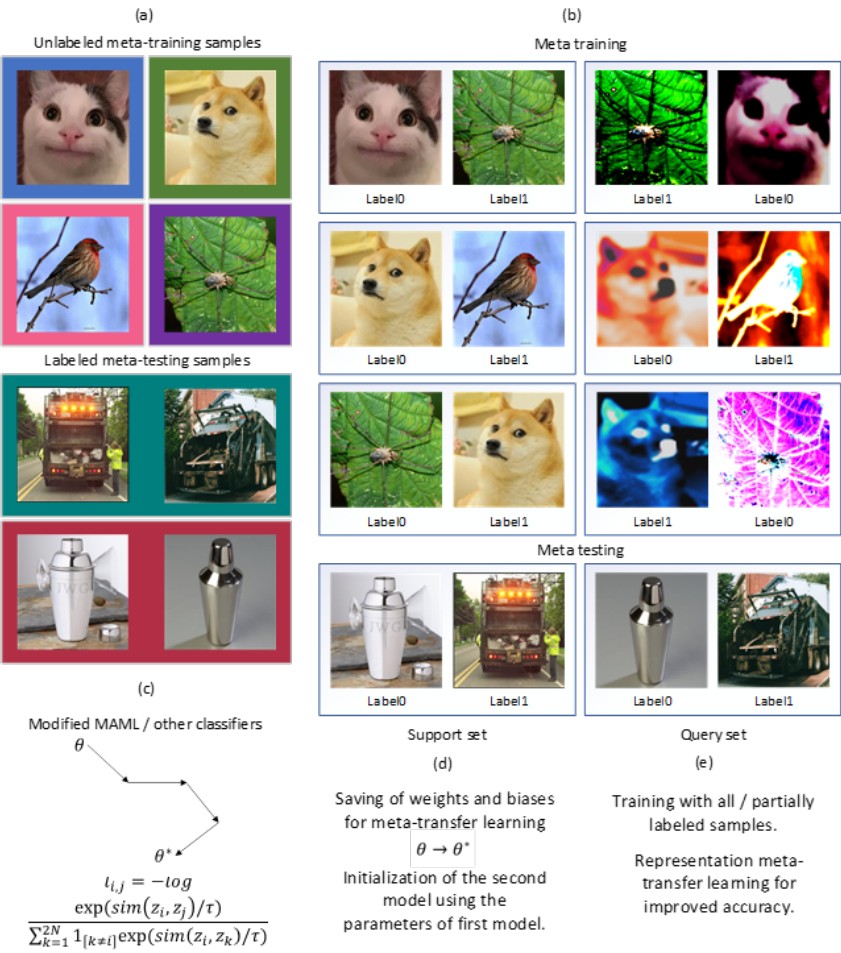

Figure 1: Steps of the proposed method (a) unlabeled samples for unsupervised learning (b) task generation for the first classifier (c) classifier for the unsupervised representation learning (d) weights and biases transfer for the supervised learning (e) supervised learning phase with initialized parameters form the first model.

---

Algorithm 1: Unsupervised representation learning for semi-supervised meta learning

**require:** unlabeled dataset, $U\{x_i\}$
**require:** $\alpha, \beta$: learning rate hyperparameters
**require:** $f(A)$: augmentation function
Initialize random parameter, $\theta$

**while** not done **do**
    generate episodes, $\{x_i\}$ and create pseudo labels, $\{y_i\}$
    **for all** $\{x_i, y_i\}$ **do**
     update inner loop of meta-learning with custom loss function or hyperparameters
    **end for**
    update outer loop of meta-learning with the regular loss function

**end while**
save weights and biases, $\theta^*$
**require:** labeled dataset, $\{x_j, y_j\} \sqsubseteq \{x_i, y_i\}$ Initialize $\theta^*$
**do** regular meta-learning steps

---

## 4 EXPERIMENTS

### 4.1 DATA AUGMENTATION FOR UNSUPERVISED REPRESENTATION LEARNING

We validate our proposed method using two different benchmark datasets in computer vision, Omniglot and mini-Imagenet. Omniglot contains images of handwritten letters from 50 different languages. This dataset is suitable for few-shot learning because it has 1623 characters or classes but only 20 instances or samples per class. We used 1200 classes for training, 100 classes for validation, and the remaining for testing. In input image dimension to the classifier is $1 \times 28 \times 28$ pixels as all are grayscale samples. On the other hand, the mini-Imagenet dataset contains $3 \times 84 \times 84$ pixels color images. It has a total of 100 classes, each with 600 samples. Here, we use 64 classes for training, 16 classes for validation, and 20 classes for testing.

Selecting the most effective data augmentation is an essential part of our research for unsupervised learning. We experimented with different augmentation methods on a trial-and-error basis and found the SimCLR augmentation with an additional augmentation gave the best output for the mini-Imagenet dataset. This section lists the results from different augmentation methods in this research. We focus on the mini-Imagenet dataset for the augmentation part because the Omniglot dataset does not require heavy data augmentation. We also try to explain why our chosen augmentation works the best for our dataset. Table 1 lists the outputs from different augmentation methods using unsupervised learning. Note that all the outputs are obtained by re-implementing different methods using our own hyperparameters, which may provide different results than other literature.

Table 1: The test accuracy (%) of unsupervised meta-learning for 5-way 1-shot (5W1S) and 20-way 1-shot (20W1S) classification using mini-Imagenet dataset. For MAML, different temperatures (denoted by $T$) are applied in the meta-inner loop.

| Augmentation method | MAML | | RN | |
| --- | --- | --- | --- | --- |
| | **5W1S** | **20W1S** | **5W1S** | **20W1S** |
| Auto augment (UMTRA*) | 30.1 ($T$=1) 35.2 ($T$=100) | 9.25 ($T$=1) 11.65 ($T$=10) | 35 | 9 |
| Resized crop + Gaussian blur + color distortions (SimCLR) | 28.4 ($T$=1) 34.4 ($T$=100) | 7.6 ($T$=1) 11.1 ($T$=10) | 32 | 7 |
| Horizontal flip(p=0.5) + color invert (p=0.5) + resized crop + Gaussian blur + color distortions (**Ours**) | 33.8 ($T$=1) **38.2** ($T$=100) | 13.65 ($T$=1) **13.95** ($T$=10) | **39** | **11.5** |

*re-implementation.

Table 2: The test accuracy (%) of unsupervised meta-learning for 5W1S and 20W1S classification using Omniglot dataset.

| Augmentation method | MAML | | RN | |
|---|---|---|---|---|
| | **5W1S** | **20W1S** | **5W1S** | **20W1S** |
| Random transformation + zero pixels (UMTRA[*]) | 48.80 | 24.94 | 61.25 | 35.78 |
| Resized crop + Gaussian blur (SimCLR) | 48.93 | 27.47 | 66.25 | 43.13 |
| Random affine transform ($30^\circ$) (Ours) | **52.83** | **27.95** | **69.12** | **44.37** |

∗re-implementation.

From Table 1, we observe the outputs from unsupervised learning for various augmentation functions. Let us discuss the accuracy of MAML first. First of all, we use the traditional meta-learning where the temperature parameter in the meta-inner loop for the SoftMax activation function is 1. A temperature of 1 means basically no temperature parameter. For MAML, we discovered that using the optimal temperature in the inner loop increased the accuracy of all the augmentation functions. It is because, when the temperature is 1, the training classifier overfits a lot due to the query set not being very challenging for the support set. When we apply the temperature, the classifier becomes less confident of the classes and can learn more features because of the introduced uncertainty. First, we apply the auto-augment function for query sample generation, which achieved slightly higher accuracy than the SimCLR augmentation function in all cases. Our augmentation function achieved 33.8% and 13.65% accuracy, which is the highest of all. We introduce temperature parameters as 100 and 10 for 5-way and 20-way, respectively. All the classifier exhibits improved accuracy for the optimal temperature, and our proposed method obtained the highest accuracy. The temperature is a hyperparameter that shows different performances for different values. In Figure 2 we illustrated the output accuracy from different temperatures to select the optimal ones. As observed, temperature 100 and 10 provides the highest accuracy for 5-way and 20-way, respectively.

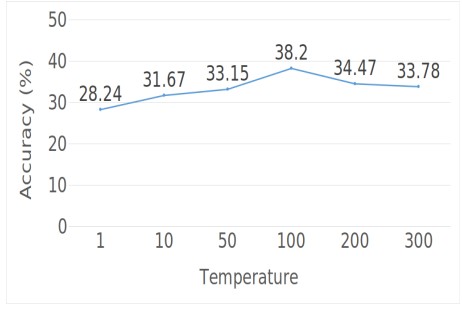

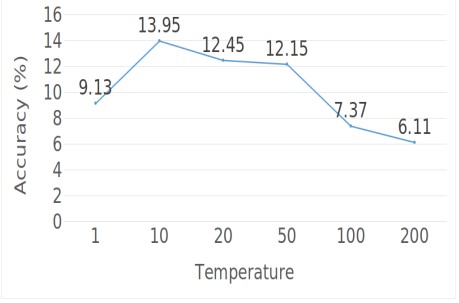

(a) 5W1S accuracy for different temperatures.      (b) 20W1S accuracy for different temperatures.

Figure 2: Effect of different temperatures on test accuracy.

For the RN, we do not modify anything in the classifier architecture; rather, our motivation is to show that the proposed method is model agnostic. Nevertheless, our combination of hyperparameters with a ResNet-18 **?** achieved higher accuracy than MAML for 5-way classification but obtained lower accuracy for 20-way classification. The auto-augment method obtained slightly higher accuracy than the SimCLR augmentation for both 5-way and 20-way classifications. However, SimCLR performs poorly for the 20-way classification and achieves only 7% accuracy. On the other hand, when we apply our proposed augmentation method, we obtain the highest accuracy for both 5-way and 20-way classifications. Nevertheless, using our proposed method, the RN module achieved higher accuracy than the MAML module for 5-way classification and lower accuracy for 20-way. Therefore, it is evident that the RN unsupervised meta-learning fails to achieve satisfactory accuracy for a higher number of classifications.

We also present the outputs from the Omniglot dataset in Table 2 to show the domain adaptability of our proposed method. In Omniglot, the support samples are quite similar to the query samples. As a result, our experiment found that doing any hard augmentation on the samples hurts the performance.

Therefore, we perform a minimum augmentation to keep the features intact and yet introduce some information in the augmented samples. Moreover, since the samples are grayscale, we could not follow the color distortion function from SimCLR. So, we only applied affine transformation within $30°$ to obtain the best transformation. This transformation distorts the samples slightly but keeps the meaning intact. In the case of SimCLR, we only apply resized crop and Gaussian blur as the color distortions do not apply to this dataset. Proof of the effectiveness of our method can be found in the experimental outputs. In this case, the SimCLR augmentation achieved higher accuracy for both MAML and RN in all 5-way 1-shot and 5-way 20-shot classifications. Our proposed method achieves the highest accuracy in all comparisons.

Our proposed method can be technically extended to an n-way 1-shot multi-query classification. Because we can generate different augmented samples in each run for multiple query generation. However, we found an accuracy drop in our proposed model when applied to multiple queries. We suspect it happens because the classifier gets overfit from multiple queries as they are not very visibly distinguishable. Table 3 represents outputs from MAML for multiple queries. The accuracy drop was significant in all 5-way 5-query and 20-way 5-query shot classifications. Therefore, we do not suggest using our method to n-query shot. One should rather apply 1-shot unsupervised learning and then transfer the learned parameters to supervised learning. We only applied a 5-way 5-query shot to RN and opted out 20-way 5-query shot because it requires substantial computational resources for a backbone of ResNet-18. We used a 12GB Nvidia 3080Ti GPU to train our MAML module. For the RN module, we used parallel computing on two 12GB Nvidia 3090Ti GPUs and Google Cloud Platform GPUs. In the Omniglot dataset, the accuracy drop for 5-way 5-shot and 20-way 5-shot were 3.68% and 0.73% for MAML and 3.87% and 5.71% for RN. For mini-Imagenet, the drops are 3.04% and 1.52% for MAML and 11% for RN (N.B. no experiments conducted for 20W1S5Q RN).

Table 3: The test accuracy (%) and drop of the proposed unsupervised meta-learning for n-way, 1-shot multi-query.

| Dataset | MAML | | RN | |
| --- | --- | --- | --- | --- |
| | 5W1S5Q | 20W1S5Q | 5W1S5Q | 20W1S5Q |
| Omniglot | 49.15 (drop 3.68) | 27.22 (drop 0.73) | 65.25 (drop 3.87) | 38.66 (drop 5.71) |
| mini-Imagenet | 32.96 (drop 3.04) | 12.43 (drop 1.52) | 28 (drop 11) | N/A |

## 4.2 SEMI-SUPERVISED META-LEARNING (SSML)

The second stage is just like regular meta-learning but initialized with the parameters from our previous method. In this section, we report our accuracy for the whole process and compare it with the traditional method. We conduct the experiments on $n$-way 1-shot 1-query and 5-shot 5-query for different classifiers. Additionally, we show how well the classifier can perform with partially labeled data instead of the whole labeled dataset.

Table 5 presents the accuracy for the original method and our proposed method for the Omniglot and mini-Imagenet datasets. We also present the outputs from the Baseline model to emphasize the effectiveness of MAML and RN. For Omniglot, our method achieves improved accuracy than the original method and proves its model-agnostic ability. In MAML, we observe significantly higher accuracy for 1-shot learning and slightly improved accuracy for 5-shot learning in both 5-way and 20-way setups. Our SSML MAML improves the accuracy of MAML further. In RN, the performance improvement is not as significant as in MAML, as the original RN already achieved very high accuracy. Nevertheless, we achieve 100% accuracy on 5W1S1Q, which improves from 99.38%. However, in both 5W5S5Q, both RN and SSML RN achieved 100% accuracy. We observe a tiny improvement for 20-way SSML MAML. In SSML RN, we observe a 4% improvement in accuracy in both 5W1S1Q and 5W5S5Q. All the outputs from MAML and RN are re-implemented in our code.

## 4.3 TRANSFERABILITY OF SSML

In this section, we test the transferability of the proposed method on different datasets. We use CIFAR-FS **?** and tieredImageNet **?** datasets for this experiment where we transfer the learned representations from miniImageNet dataset. The CIFAR-FS dataset has 100 classes and 600 images per class. Train, test validation sets are split into 64, 16 and 20, respectively. The tieredImageNet

consists of 608 classes and 779,165 total images. We use 351 classes for training, 97 for validation and 160 for testing.

We initialize SSML MAML with miniImageNet representation and fine-tune on both datasets. The outputs are listed in Table 4. In all cases, SSML MAML improves accuracy over MAML. The most significant improvement is for CIFAR-FS 5W1S1Q, which is 3.6%. This proves that the proposed method can also transfer the learned representations to different domains for improved accuracy.

Table 4: Transferablity of SSML MAML for different datasets.

| Data | Method | 5W1S1Q | 5W5S5Q | 20W1S1Q | 20W5S5Q |
| --- | --- | --- | --- | --- | --- |
| CIFAR-FS | MAML | 49.6 | 71.2 | 25.76 | 42.16 |
| | SSML MAML | 53.2 | 71.73 | 26.34 | 42.8 |
| tieredImageNet | MAML | 48 | 61.47 | 19.56 | 32.35 |
| | SSML MAML | 48.2 | 62.04 | 20.1 | 33.23 |

## 5 CONCLUSION

In this research, we propose a meta-learning strategy that learns the latent representation from the dataset using unsupervised meta-learning and then performs SSML using the learned parameters. Unsupervised learning gives a performance boost to supervised learning. Therefore, our method is fast adaptive and obtains improved accuracy. Our unsupervised method depends on effective data augmentation for query sample generation. Additionally, we visually represent why our proposed combination of augmentations is more effective than other augmentations. The temperature-scaled SoftMax also plays a vital role in unsupervised classification accuracy. We tested our proposed model with two different datasets and models. Our method achieve better test accuracy in all cases than the original methods. We also show that our method can retain good accuracy and lower loss when trained on partially labeled training samples.

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

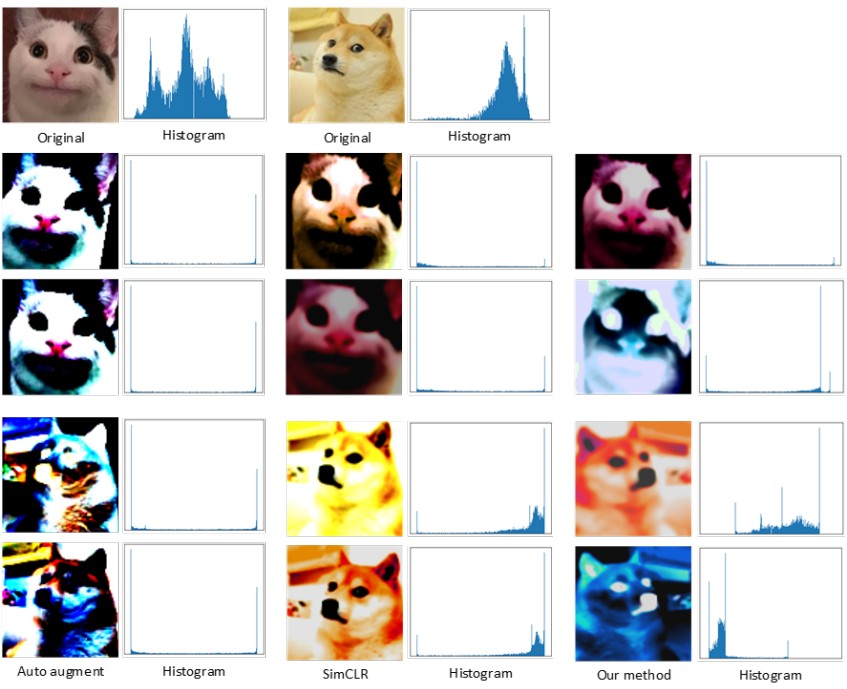

Figure 3: Histogram of pixel intensities for different augmentation methods.

## A  WHY OUR METHOD IS EFFECTIVE

In Figure 3, we explained why our proposed method works effectively by visualizing the pixel intensities from the histograms of augmented images. The histogram analysis shows how much uncertainty is introduced in different augmented samples compared to the original samples. In both auto augment and SimCLR augmentation, we find the histogram of augmented samples are very similar to each other. That means all the augmented samples fail to introduce enough new uncertainties in each augmentation. It is essential because a sample can appear often in meta-learning in different episodes. So, we must supply query samples with distinguishable features in each run. On the other hand, for the proposed augmentation method, we find the histograms have whole new pixel intensities for each run. Therefore, the features have new information in each query sample. It can also be explained by the uncertainty we introduce in our augmentation function by doing a horizontal flip and color invert with a 50% probability for each one. Therefore, our proposed method achieves the highest accuracy for unsupervised meta-learning learning.

## B  ABLATION STUDY

Additionally, we highlight that our method can obtain high accuracy or less accuracy loss for partially labeled datasets (Table 6). We test our hypothesis on mini-Imagenet only because it contains 600 samples per class, whereas Omniglot only has 20 samples per class. We randomly select 50% (300) and 25% (150) training samples from the mini-Imagenet data and train our classifier to compare the proposed and original methods. This time we report the percentage accuracy drop from the main output (trained on 100% samples) to have a fair comparison between the original method and SSML. It is obtained as $((all\,labeled\,accuracy - partially\,labeled\,accuracy)/all\,labeled\,accuracy) \times 100\%$. In most outputs, our proposed method has less drop except for SSML RN with 50% and 25% labeled data for 5W5S5Q and 5W1S1Q, respectively. In MAML and SSML MAML, for 5W1S1Q, we have negative accuracy drop percentages. This is because the accuracy, in fact, increases when we train MAML with 50% data in this setup. We hypothesize this improvement is due to the episode generation with fewer samples in each class. Some research points out that having a large number of meta-training data can counter-intuitively hurt performance. Because of multiple possibilities for generating each episode, the probability of all the samples appearing in the episodes will be lower.

402 For example, Triantafillou et al. **?** found that having a large meta-dataset hurts the accuracy of the
403 mini-Imagenet dataset. Setlur et al. **?** showed that having a fixed support set and having less diversity
404 can improve accuracy. This new research direction deals with the optimal number of samples in
405 meta-training and a more effective way of generating the episodes. We aim to focus on this area in
406 our future research.

Table 5: The test accuracy (%) of the supervised meta-learning for the Omniglot dataset.

| Method | Omniglot | | | | mini-Imagenet | | | |
|---|---|---|---|---|---|---|---|---|
| | 5-way accuracy | | 20-way accuracy | | 5-way accuracy | | 20-way accuracy | |
| | 1S1Q | 5S5Q | 1S1Q | 5S5Q | 1S1Q | 5S5Q | 1S1Q | 5S5Q |
| Baseline | 86 | 97.6 | 72.9 | 92.3 | 38.4 | 51.2 | N/A | N/A |
| MAML* | 93.8 | 98.3 | 82.5 | 92.3 | 46.8 | 61.6 | 18.75 | 30.4 |
| RN* | 99.38 | 100 | 97.19 | 99.59 | 53 | 64 | 24.25 | N/A |
| SSML MAML (**Ours**) | **96.44** | **98.34** | **83.35** | **92.72** | **47.6** | **61.8** | **18.88** | **30.71** |
| SSML RN (**Ours**) | **100** | **100** | **97.34** | **99.69** | **57** | **67** | **25** | **N/A** |

∗re-implementation.

Table 6: Accuracy drop (%) of supervised meta-learning for the partially labeled mini-Imagenet training set.

| Method | 5-way accuracy | | | | 20-way accuracy | | | |
|---|---|---|---|---|---|---|---|---|
| | 1S1Q | % Drop | 5S5Q | % Drop | 1S1Q | % Drop | 5S5Q | % Drop |
| MAML (50% labeled data)* | 48.2 | -2.99 | 61.45 | 1.87 | 18.75 | 5.07 | 28.42 | 6.51 |
| SSML MAML (50% labeled data) | 49.4 | **-3.78** | 61.04 | **1.23** | 17.94 | **4.98** | 28.83 | **6.12** |
| MAML (25% labeled data)* | 46 | 1.71 | 57.4 | 6.82 | 16.25 | 13.33 | 26.54 | 12.70 |
| SSML MAML (25% labeled data) | 47.2 | **0.84** | 58.25 | **5.74** | 17.5 | **7.31** | 27.05 | **11.92** |
| RN (50% labeled data)* | 39 | 26.42 | 58.2 | **9.06** | 19.75 | 18.56 | N/A | N/A |
| SSML RN (50% labeled data) | 43 | **24.56** | 59.2 | 11.64 | 22.25 | **11** | N/A | N/A |
| RN (25% labeled data)* | 38 | **28.3** | 51.8 | 19.06 | 18.75 | 22.68 | N/A | N/A |
| SSML RN (25% labeled data) | 40 | 29.82 | 54.4 | **18.81** | 20.25 | **19** | N/A | N/A |

∗re-implementation.

