# OpenReview forum: "Unsupervised Representation Learning to Aid Semi-Supervised Meta Learning"
_ICLR.cc/2024/Conference — ICLR 2024 Conference Withdrawn Submission_

### Official Review · Reviewer_vJfx · 2023-10-25

**Soundness:** 1 poor
**Presentation:** 1 poor
**Contribution:** 1 poor
**Rating:** 3
**Confidence:** 4

**Summary:**

This paper proposes solve the meta-learning problem by taking augmented data samples as the query set during training phase. Generally, the writing of paper is quite poor, the proposed method is filled with subjective conjectures and confusing motivations. I do not think that the proposed method makes sense.

**Strengths:**

- The experimental results looks good.

**Weaknesses:**

__Main concerns:__
- __[Poor writing]__ The writing of this paper is poor. For example, the format of references are not well managed. More efforts are required to improve the quality of the paper.

- __[Unclear formulation]__ The problem studied in this paper is not well formulated, and the notation used in this paper is quite confusing.

- __[Farfetched statements]__ There are many confusing statements that are confusing and farfetched in this paper.

- __[Inconvincing results]__ Due to the confusing method settings, I do not think the empirical results are convincing.


__Minor concerns:__
- __[Limited Novelty]__ The proposed method is not novel.

**Questions:**

1. (Line 44.) Why large datasets where the probability of drawing samples from the same class is much lower helps achieve better performance?

2. (Line 119.) I do not think that $x_{i, j}$ is an appropriate way to represent data samples. Besides, we usually use the upper case to denote the variables, it is a little bit confusing to use it to represent data pool. Meanwhile, if $X_N$ denotes the unlabelled samples, how to sample $n$-way $1$-shot tasks?

3. (Line 121.) What does the sentence "We only design n-way (n is the number of ways or classes), 1-shot support
122 sets because each sample in the support set is drawn randomly, and we cannot randomly add more
123 same-class support samples to that set" mean? In meta-learning settings, $N$-way $K$-shot tasks, such as 5-way 5-shot tasks, are common.

4. (Line 127.) What does $y_{i, j}$ mean? What the function of so-called "labeled values"? Do you mean the random assigned class numbers?

5. What does the notation $f(A)$ mean? What is $A$?

6. In my opinion, the query set in meta-learning mean the data belonging to the same classes as support data but are not observed by model during adaptation phase (inner loop). So I suggest that use other term to describe the augmented support data to make it clear.

7. In Line 128, $c$ means the number of ways, but it is defined further in Line 136 as the total number of classes. So, what is the structure of the tasks?

8. Line 134. What does the sentence "We intuitively know that when we draw a few samples from a large pool of data, more than one sample belonging to the same class is low" mean?

9. In Line 139-140, the authors claim that Omniglot has less classes than mini-ImageNet. In fact, mini-ImageNet only owns 100 classes with 600 samples in each class. Besides, why is it more likely that all drawn samples will originate from different classes for datasets that have less number of samples in each class? In fact, I do not understand the generation process of support-query set described in Line 134-146.

10. Sometimes augmentations tend to modify the information in images, so how do these techniques help improve the performance?

11. What are the evidence for your statement that the classifier of MAML trained on RGB samples has a severe overfitting issue ...... Therefore, the classifier learns very little during the training phase? Such statement is farfetched.

12. The expressions of $\theta_i^{'}$ and $\theta^i$ are inconsistent.

13. Where is the semi-supervised meta-learning algorithm? What the goal of section 3.2?


__Small problems:__
1. Line 139. It would be better to replace "prior" with "former".

---

### Official Review · Reviewer_SMhQ · 2023-10-30

**Soundness:** 1 poor
**Presentation:** 1 poor
**Contribution:** 1 poor
**Rating:** 3
**Confidence:** 3

**Summary:**

This paper proposes a data augmentation methods to generate query set inspired from SimCLR, and uses some other techniques (such as, temperature-scaling, etc) to help the semi-supervised meta learning. Some experiments are conducted to help explain the effectiveness of the proposed approach.

**Strengths:**

I did not find the strengths.

**Weaknesses:**

1. The motivation is not clear.
2. The novelty of the proposed approach is limited. It seems all techniques are common methods used in the ML community.
3. The representation is not good and needs really really careful writing to improve.
4. The experiments are not extensive.

**Questions:**

No questions.

---

### Official Review · Reviewer_8qJe · 2023-11-01

**Soundness:** 1 poor
**Presentation:** 1 poor
**Contribution:** 1 poor
**Rating:** 1
**Confidence:** 4

**Summary:**

They propose the data augmentation technique that has more synergy with simCLR. To avoid the overfitting issue of meta-learning, they utilize the temperature scaling. Lastly, they replace random initialization of meta-learning with unsupervised representation learning, and then they train the model in supervised manner.

**Strengths:**

I don't think that this paper has some contributions.

**Weaknesses:**

**1 (Novelty).** To the best of my knowledge, I do not catch the novelty of this paper. They claim that data augmentation that they proposed is crucial for performance enhancement by combining with SimCLR, but the data augmentation is not new (just use AutoAugment). Moreover, they point out that they utilize the temperature scaling while meta-training as a their novelty, but this is a general technique for training the models. For methodology perspective, lastly, they propose two-stage training method (unsupervised representation learning -> supervised learning), there are nothing special for using those methods.

**2 (Experiment).** In experiment parts, I think it is not enough to validate the effectiveness of their proposed method. First, the authors should compare proposed method with not only simCLR but also recent meta-learning techniques [1]. Furthermore, they should report the ablation studies. For example, the authors can study the effectiveness of their each training step, so we can figure it out which step is more important for performance enhancement.

Besides those weakness points, there are lots of things to enhance, but I stop to write the comments.

**Questions:**

**1 (Citation Error).** I think the paper has citation errors, so it is hard to know that which papers can be cited. It should be fixed.

---

### Official Review · Reviewer_fvcV · 2023-11-01

**Soundness:** 1 poor
**Presentation:** 2 fair
**Contribution:** 1 poor
**Rating:** 3
**Confidence:** 4

**Summary:**

The paper proposes a one-hot unsupervised meta-learning to learn the latent representation of the training samples to improve semi-supervised meta-learning. A set of data augmentation is applied to obtain a query set for assigning probability on different classes, and a temperature-scale cross-entropy loss is introduced to reduce overfitting. Further, existing meta-learning methods model agnostic meta-learning (MAML) and Relation Network (RN) are applied for training. The proposed approach is evaluated on the Omniglot and mini-Imagenet datasets and shown to be better than the existing method MAML.

**Strengths:**

The paper is easy to follow and understand. The implementation and technical details are mostly clear.

**Weaknesses:**

The contribution of this paper does not meet the standard for publishing at ICLR. First, the novelty of this work is limited. The proposed method appears to a combination of existing proposed methods such as model agnostic meta-learning (MAML) and Relation Network (RN). It is hard to pick up what are the new ideas that contribute to this community. Second, the evaluation of the proposed method is weak and not convincing. The proposed method only compared to the MAML approach which is an existing meta-learning approach published in ICML2017. Third, the presentation and written quality of this work is weak, e.g., the citations are all question marks, and presented figures are unclear to follow the key ideas.

**Questions:**

Why the combination of all the different existing techniques are important to solve semi-supervised meta-learning? Please justify with written motivation and empirical experimental results.

**Details Of Ethics Concerns:**

No concern on ethics.